# Smokers’ Self-Report and Behavioral Reactivity to Combined Personal Smoking Cues (Proximal + Environment + People): A Pilot Study

**DOI:** 10.3390/brainsci12111547

**Published:** 2022-11-15

**Authors:** Cynthia A. Conklin, Brian A. Coffman, F. Joseph McClernon, Christopher Joyce

**Affiliations:** 1Department of Psychiatry, University of Pittsburgh, Pittsburgh, PA 15213, USA; 2Psychiatry & Behavioral Sciences, Duke University, Durham, NC 27705, USA

**Keywords:** smoking, cue reactivity, cue exposure, smoking behavior

## Abstract

Cue reactivity (CR) among smokers exposed to smoking-related stimuli, both proximal (e.g., cigarettes, lighter) and distal (environments, people), has been well-demonstrated. Furthermore, past work has shown that combining proximal smoking cues with smoking environment cues increases cue-provoked craving and smoking behavior above that elicited by either cue type alone. In this pilot study, we examined the impact of combining three personal cues, proximal + environment + people, on subjective and behavioral cue reactivity among smokers. To further understand the impact of this method, we also tested reactivity under the conditions of both smoking satiety and deprivation. In addition, we examined the extent to which cue-induced craving predicted immediate subsequent smoking. Fifteen smokers completed six sessions, of which two focused on the intake and development of personal cues and four involved personal cue reactivity sessions: (1) deprived, smoking cue combination, (2) deprived, nonsmoking cue combination, (3) sated, smoking combination, and (4) sated, nonsmoking cue combination. Cue-provoked craving was greater and smokers were quicker to light a cigarette and smoked more during their exposure to smoking rather than nonsmoking cues and in deprived compared to sated conditions, with no interaction between these variables. While deprived, greater cue-provoked craving in response to smoking cues was correlated with a quicker latency to light a cigarette. This work supports the feasibility of presenting three personal smoking-related combinations of cues within a cue reactivity paradigm and highlights the robust reactivity that this methodology can evoke in smokers.

## 1. Introduction

Smoking cue reactivity (CR) research [1], in which individuals are presented with smoking-related and neutral stimuli to assess their subjective, physiological, and/or behavioral responses to such cues, has established that exposure to smoking cues can evoke strong craving [1,2,3], and an increase in, actual smoking [4]. Moreover, smokers readily report that confrontation with smoking cues (e.g., seeing a cigarette) is a vital aspect of their difficulty or inability to stay quit [5]. Research has consistently shown the robust reactivity of smokers exposed to proximal smoking cues, most of which are closely linked to actual drug administration (e.g., cigarettes, lighter) [6]. However, with the advent of new technology (e.g., VI, digital cameras, smartphones), more cue studies are incorporating distal smoking cues, which are cues less directly tied to the act of smoking [6], such as environments and people [7,8,9,10,11,12,13,14,15]. This is a crucial aspect of CR smoking research, as our past work has shown that the personal environments and people in a smoker’s own life can, even when proximal cues are not present, trigger equivalently strong cue-induced craving [8,16,17], changes in mood [8], brain reactivity [12,18], and increases in actual smoking [12,13,17,19].

Given the array of cues that can induce strong cravings to smoke and motivate actual smoking, CR research needs to incorporate multiple cue types to offer a complete picture of the impacts of cues on smoking [20,21]. Likewise, the limited focus within cue-based addiction treatments on reducing reactivity to proximal cues alone is a purported culprit responsible for the limited efficacy of cue exposure treatments (CETs), which aim to diminish reactivity to smoking cues through unreinforced cue exposure [14,20,22]. CETs may benefit greatly from the inclusion of exposure among the actual places and people a smoker is most likely to encounter outside of treatment and thereby target more relevant personal cues that trigger smokers’ strongest urges to smoke [18,21]. Thus, proximal and personal distal cues need to be included in CR and CET investigations to maximally evoke craving and gain a more complete understanding of how cues function and to discover new methods that can be used to attenuate smokers’ robust reactivity to cues.

Gaining a more complete picture of how smoking cues function requires researchers to not only study various cue types but also examine them as they likely occur in the real world, namely, in combination. For example, smokers may find themselves in a smoking-permissive environment while interacting with a friend who also smokes. Our past work has shown that combining smoking cues arising in personal smoking environments with proximal smoking cues leads not only to higher levels of self-reported craving but also to increases in actual smoking compared to single cues or smoking cue + nonsmoking cue combinations [17]. Similarly, other researchers have identified enhanced reactivity to smoking cues when they are presented within a smoking context (e.g., pictured in a pub) [15]. Ideally, presenting personal proximal, environment, and people cues in combination should lead to highly relevant and salient smoking scenarios, allowing for the investigation of robust reactivity. The feasibility and impact of performing this research have yet to be investigated. Additionally, cue combinations need to be investigated under the conditions of deprivation and satiety [23,24]. Although past CR studies have rarely revealed interactive effects between deprivation and cue-induced craving (i.e., deprived smokers show additive, not interactive, elevations in cue-provoked craving and smoking behavior compared to non-deprived smokers), more complex personal cue presentations might significantly alter or enhance reactivity, and interactions may be revealed.

As an initial step in advancing our multi-cue presentation methodology, the goals of the present pilot study were defined as two-fold. The first goal was to present a combination of three personal cue-types (proximal, environment, and people) within a cue reactivity paradigm and examine the magnitude of self-reported craving and smoking behavior (latency to smoke and puff volume) in response to combinations of personal smoking cues versus personal nonsmoking cues. The second was to determine the impact of deprivation on reactivity to these combined cue exposures. We hypothesized that combined smoking cues would increase self-reported craving and the smoking topography compared to combined nonsmoking cues and sought to examine if deprivation would increase self-report craving and smoking topography compared to non-deprivation. In line with our past findings, we also hypothesized a correlation between cue-provoked craving and smoking topography measures, such that higher cue-induced craving would be associated with a shorter latency to light and greater total puff volume.

## 2. Materials and Methods

### 2.1. Participants

Fifteen daily smokers (8 men and 7 women) were recruited using advertisements and flyers requesting, “healthy men and women smokers, ages 18–65 [to participate in] a research study investigating smoking cues”. The eligible participants were daily non-quitting smokers of 18–65 years old (M = 37.3; SD = 12.2; range = 19–51), who had smoked 10 or more cigarettes/day for at least the past year (M = 16.6; SD = 6.37; range = 10–35) and had a carbon monoxide concentration (CO) above 8 ppm (M = 25.9; SD = 14.9; range = 10–59). This range of cigarettes per day reflected the average smoking in the U.S. at the time of the study [25]. The CO cutoff was chosen to ensure that regular daily smokers entered the study [26]. At screening, potential participants were told that the current study did not involve treatment. Those who were seeking smoking cessation treatment were offered referral information. Participants had an average Fagerström test of nicotine dependence (FTND) [27] score of 5.07 (SD 1.90; range 2–8) and were paid USD 200 for completing the 6-session study.

### 2.2. Study Overview

The study comprised 6 sessions. Session 1 focused on a demographic assessment and a cue interview conducted to determine the pictures that participants would take with a borrowed camera to create personal environment and people cues. Session 2 was a brief camera drop-off session to review the pictures and pair the people and environment cues. This session also involved participants smoking one cigarette with the CReSS smoking device to train them for in-session smoking during the later cue reactivity sessions (3–6). Sessions 3–6 were cue reactivity sessions, each conducted on a separate day, during which one of 4 conditions was completed: (1) deprived—smoking cue combinations, (2) deprived—nonsmoking cue combinations, (3) sated—smoking cue combinations, and (4) sated—nonsmoking cue combinations. Each cue reactivity session involved 4 combined cue trials, each followed by self-report craving ratings and then a 12 min ad lib smoking period, during which the cue combinations for that session were repeatedly presented (See Figure 1).

### 2.3. Session 1

During this 90 min session, participants signed informed consent, stated the time since they last smoked, and provided a carbon monoxide (CO) expired air sample using a Vitalograph CO monitor (Vitalograph; Lenexa, KS, USA). They then completed the Smoking History Form, which included the FTND [27]. Using a semi-structured interview developed and validated in our past work [16,17], the experimenter determined the top personal smoking and nonsmoking people and environments of which each participant would take pictures over the next week. The participants then photographed 4 smoking and 4 nonsmoking places that they visited at least weekly from 4 angles each (two approaching the environment and two from within it). They were told to not include any people or proximal cues in the photographs. The participants also took separate pictures of 4 people around whom they smoke and 4 around whom they do not in their weekly lives. These photographs were taken of each individual from the shoulders up, with a neutral facial expression. After determining the environments and people to be photographed, the participant underwent picture-taking training, with the experiment room serving as a sample environment and the experimenter as an example person. Prior to leaving, the participants were given a FujiFilm Finepix J38 digital camera to borrow (FujiFilm Co.; Edison, NJ, 08837, USA), a written reminder card listing the environments and people of which/whom they should take pictures, and the date/time of their next session. Additionally, all the people of whom pictures were taken signed a consent form agreeing to have their photograph used in an experimental study.

### 2.4. Session 2

A brief second session was conducted approximately one week after the initial session. This gave the participants time to take pictures of the environments and people they identified in Session 1. The participants again provided an expired air CO sample to capture their mid-day exposure to smoking. They then dropped off the camera containing their photographs and confirmed which people were paired with which environments. The participant then smoked a cigarette using the CReSS smoking topography device (Borgwaldt KC GmbH, Hamburg, Germany). This served to train the participants in how to use the CReSS device to smoke in future sessions. The experimenter then scheduled the participant’s third session.

### 2.5. Sessions 3–6

Four cue reactivity sessions, during which the participants viewed combinations of their smoking or nonsmoking environments, people, and proximal cues, were conducted on four separate days. Each session was approximately 2 h long, with the first occurring approximately 1 week after session 2. During the interim week, the experimenters edited the participants’ photographs and inserted them into the cue presentation program. The photo editing was achieved using Adobe^®^ Photoshop^®^ software to eliminate any unwanted stimuli (i.e., proximal smoking cues, people, or alcohol) that appeared in the participants’ environment pictures using the spot-healing and clone-stamping functions in Adobe^®^ Photoshop^®^. The people pictures were cropped equivalently (to the top of the head and top of the shoulders) on a neutral cream-colored background. Sessions 3–6 occurred within 3 days of each other. Each session included either all the smoking or all the nonsmoking cue combinations of the proximal + environment + people cues under the condition of either smoking satiety or 12 h of deprivation. The order of sessions was equally counterbalanced across subjects. Separate sessions for each cue combination/deprivation condition were used to avoid the cross-over effects of ad lib smoking during exposure to each cue combination type, as well as to control for the session order.

Upon arrival at each session, a CO sample was taken. For the two sated sessions, the participant then smoked one cigarette using the CReSS system, while in the deprived sessions, they did not smoke. The participants were provided with an overview of the session, which included instructions about viewing the cues and completing the post-trial ratings (see details below). The people cues were displayed on a 22 in. monitor (ViewSonic Corporation; Walnut, CA, USA) in front and to the left of the participant, with the paired environment pictures for that person presented on a project wall (8 × 15 ft.) using an InFocus DLP short-throw projector (InFocus Corporation; Portland, OR, USA). The proximal cues were set on a small table in front and to the right of the participant. The smoking proximal cue was a cigarette of the participant’s preferred brand sitting in an ashtray, with a lighter next to it. The nonsmoking cue was a pencil sitting on a small pad of paper with a red eraser next to it. These methods of cue presentation were used to present stimuli as they would likely occur in the real world, with both the people and environments in appropriate proximity to the participant and of a near-life size (see Figure 2).

The participant first completed one practice trial to confirm that he/she understood the automated cue reactivity procedure. The experimenter then exited the room, and the 4 CR picture trials began. Each picture trial followed the same format: 20 s relaxation and 40 s picture viewing (10 s for each of the 4 environment angles within a picture set, with the corresponding people and proximal cues present), followed by a screen instructing the participant to complete the post-trial craving ratings (the 4-item QSU) using a mouse and answering the questions as they appeared on the small computer on which the people cue was presented. The pictorial stimuli were accurately timed so that the environment and people cues were presented on the two separate screens in synch. The program also controlled the post-trial ratings. The instructions to the participants read, in part:


*A prompt will appear on the screen instructing you to sit back in the chair and relax. Following that, pictures of people will appear on the computer screen to the left in front of you, objects are on the table to the right in front of you, and pictures of environments will be displayed on the wall across from you. You are to focus on the people and items within the environments you see. Keep focusing on these things until the computer screen changes and prompts you to answer questions based on how you felt while focusing on that scenario.*


Pictorial cues were not present during the post-trial ratings. After the last of the 4 trials, the participant saw a screen indicating:


*When the pictures reappear, you may smoke as much or as little as you like. You do not have to smoke if you choose not to. If you smoke, you must light the cigarette and use the cigarette holder the same way you did earlier.*


The participant’s cigarettes, the CReSS smoking device, one lighter, and an ashtray were on a tray covered by a small box on a side table near the participant, but they were out of view. A second screen then appeared instructing the participant to move the tray forward, uncover it, place a cigarette in the holder, place it back on the tray, and click the mouse to advance to the next screen. This last mouse click began the smoking latency timer. The environment and people–picture pairs, which the participant saw in the previous 4 exposure trials, were then presented and repeated randomly for 12-min, during which he/she could engage in ad lib smoking. Behavioral smoking indices of latency to light and the puff volume were collected during this time. The latency to light was entered as the maximum duration (12 min) in cases where the participants did not smoke during the ad lib period. Following the ad lib period, the participant was scheduled for his/her next session. If it was the last session, the participant was debriefed and paid.

### 2.6. Data Analysis

The dependent variables (craving, latency to light, and total puff volume) were first checked for missing values, and their agreement with assumptions of the ANOVA was examined. After applying a log transformation to the latency to light variables, all the variables were normally distributed, with a skewedness of <1.5 and kurtosis pf < 4. The relationships between the group demographics (FTND, cigarettes per day, sex) and dependent variables were examined using Person correlations and *t*-tests where appropriate. We used univariate repeated-measure RM-ANOVA to assess the within-subject effects of the cue type (smoking vs. nonsmoking), abstinence (deprived vs. sated), and their interaction on the cue-provoked craving scores. As the latency to light and total puff volume were moderately correlated for each cue type and the abstinence condition cell (*r*’s = −0.83–−0.26), multivariate RM-ANOVA (RM-MANOVA) was used to assess the effects of these independent variables on the smoking topography measures. The analysis of the significant multivariate effects were followed by univariate RM-ANOVA. We used Pearson’s correlation to assess the bivariate relationships between cue-provoked craving and the smoking topography measures (latency to light/total puff volume), with Bonferroni-corrected α = 0.025. Cohen’s *d* is given as a measure of the effect size.

## 3. Results

As predicted, the cue-provoked craving scores were greater for the smoking cues than the nonsmoking cues (*F*_(1,14)_ = 18.0; *p* < 0.001; *d* = 0.88) and for the deprived compared to sated conditions (*F*_(1,14)_ = 37.2; *p* < 0.001; *d* = 1.74), with large effect sizes and no interaction between these variables (*p* > 0.1). Similarly, RM-MANOVA revealed the significant main effects of the cue type (λ = 0.60; *F*_(2,14)_ = 4.7; *p* < 0.05) and abstinence (λ = 0.17; *F*_(2,14)_ = 34.2; *p* < 0.001) on the smoking topography. Further investigation of these effects by RM-ANOVA revealed that latency to light occurred earlier for the smoking cues than the nonsmoking cues (*F*_(1,15)_ = 5.4; *p* < 0.05; *d* = 0.60) and for the deprived compared to sated conditions (*F*_(1,15)_ = 63.5; *p* < 0.001; *d* = 2.06). Additionally, the total puff volume was greater for the smoking cues than the nonsmoking cues (*F*_(1,15)_ = 9.7; *p* < 0.01; *d* = 0.60) and for the deprived compared to sated conditions (*F*_(1,15)_ = 28.2; *p* < 0.001; *d* = 1.24). Smoking-cue-provoked craving while deprived was negatively correlated with latency to light (*r =* −0.57; *p* < 0.025) but not the total puff volume (*r* = 0.13; n.s.; Figure 3). These correlations were not significant in the sated condition (|*r*’s| < 0.3). Furthermore, the difference between these correlation statistics was statistically significant (*z* = 1.9; *p* < 0.5). No group demographics (FTND, cigarettes per day, or sex) were correlated with the dependent variables. Descriptive statistics are given in Table 1.

## 4. Discussion

As anticipated, the combinations of personal smoking-related cues incorporating proximal, environment, and people cues from an individual’s own life led to greater cue-induced craving, faster latency to light a cigarette, and a greater overall puff volume compared to the nonsmoking cue combination. All of the effect sizes of these findings can be considered large by Cohen’s standards [28], a finding in line with our past work examining proximal and distal cues to smoke [8,9]. Deprivation increased the overall craving levels but did not interact with cue-induced craving. Furthermore, as revealed in past studies [17], there was some evidence to suggest that the magnitude of cue-induced craving was correlated with immediate subsequent smoking. Here, the greater the craving that the cues evoked during deprivation was, the faster the smokers lit up.

Cue reactivity is a well-established phenomenon [2,29]. However, by advancing the methods to more completely capture individuals’ real-world scenarios, we can provide a useful methodology for studying key underlying mechanisms of smoking maintenance and relapse, namely, cue provoked-craving and smoking behavior. The combination of cues has the benefit of capturing multiple features that are likely to occur in concert in the real world, thus recreating cue-rich scenarios that smokers encounter in their daily lives. As noted in even the earliest conditioning studies [30,31], the closer the cues come to capturing the scenarios of original learning (i.e., the situations in which smoking has repeatedly occurred), the greater the reactivity they should provoke. Subsequently, in order to reduce cue responding, it is imperative to elicit a strong reactivity to then extinguish [32]. Our prior work suggests that the personalization of cues can achieve these goals more effectively than standard smoking and nonsmoking cues [16]. Furthermore, we found that combining two smoking cues, the proximal and environment, led to a stronger reactivity than either alone [17]. Our goal here was to assess the feasibility and impact of combining three personal cue types (proximal + environment + people) within a cue reactivity paradigm, a method that led to strong cue-provoked reactivity.

The potential utility of this method for future investigations is evident. In-lab presentations of cue-provoking scenarios taken from smokers’ lives affords us a better proxy to indicate how efficacious a new treatment might be in combatting real-world craving. This is achieved while avoiding the inherent difficulties of real-world exposure, such as cue avoidance. It also allows for a strong cue responding base from which we can systematically test novel treatment methods aimed at reducing cue-provoked reactivity. Interest in reducing cue-provoked craving has led to investigations specifically examining the therapeutic efficacy of cessation medications and other compounds, such as naltrexone [33], nicotine replacement (NRT) [34], and varenicline [35], as well as new brain stimulation methods, such as transcranial direct current stimulation (tDCS) [36,37], and cognitive tasks, such as approach/avoidance cue training (AAT) [38,39]. The efficacy of these treatments in the context of real-world smoking may be enhanced by examining their effects in a lab simulation of conditions similar to those encountered outside of treatment. Similarly, changes in reactivity to strong cue presentations in the lab might provide better methods of tracking treatment progress [40] or offer predictions about who is likely to fair well with certain cessation therapies [41,42]. The results also suggest an anticipated additive effect of deprivation on cue-induced craving and smoking topography but no evidence of interaction. This is in line with past findings, suggesting that deprivation increases the underlying craving (or tonic craving) but not cue-provoked craving, per se [23]. Exposure to cues can promote craving even when smokers are sated and enhance it further when they are deprived. Some researchers have expressed their concern that underlying craving prior to cue exposure may lead to ceiling effects in craving that dilute the perceived impacts of cues [43]. Although additively enhanced during deprivation, patterns of cue-induced craving and smoking behavior are similar whether the individual is deprived or smoking as usual. However, 2 of the 15 participants reported nondifferential craving in response to smoking and nonsmoking cues during deprivation and rated their craving during exposure to both at the very top of the scale (i.e., 100). This suggests that, under deprived conditions, tonic craving might shield or override the impacts of cues on craving among some smokers, and/or that craving is so high in the deprived condition that the scale (0–100) is inadequate for assessing the additional craving evoked by cues.

The goal of the present pilot study was limited in scope, and the sample size was small, precluding a deeper understanding of the impacts of multi-cue presentations. Combining three personal cue types within a cue reactivity paradigm was methodologically successful but limited due to the inclusion of only combined smoking cues compared to combined nonsmoking cues, thus disenabling an investigation of how each smoking cue type individually affected smokers or how smoking cues combined with nonsmoking cues might work in concert to impact craving and smoking behavior. For example, if an individual has cigarettes while sitting next to a smoking friend but is in church, his/her craving might be high regardless of the lower likelihood of actually smoking. Our past study combining proximal and environment cues [17] revealed that two smoking-related cues evoked greater craving and smoking behavior, both of which decreased when a nonsmoking cue was presented with a smoking cue, with the lowest reactivity revealed when both cues were nonsmoking cues. Given the large effect size (*d* = 1.08) of the difference in self-reported cue-provoked craving in response to dual smoking cues versus dual neutral cues (proximal + environment) in that study, we included only a small sample in this pilot. Specifically, we projected having 97.3% power to detect an effect on cue-provoked craving with 15 subjects and found multiple significant effects. However, a large-scale study of multi-cues is required to assess the impacts of the three cue types included here in various combinations. Future research, possibly that using eye-tracking methods, which has been performed in several cue-related studies [44,45], might investigate where the subject’s attention is drawn when multi-cues are presented.

Lastly, we found no impact of sex or smoking level on cue reactivity. Although an absence of sex differences is in line with our past smoking cue reactivity findings [8,9,16,17,42] and those of other laboratories [11,33,46], the evidence of sex differences across smoking cue reactivity studies is mixed. Some studies have found higher cue-provoked craving in women compared to men [47,48,49,50], and brain reactivity smoking cue studies have revealed differences between male and female smokers [50,51,52]. As noted by other researchers [53], variations in cues types, cue manipulations, and the outcomes assessed likely explain the variability in findings of sex differences. Thus far, our personal pictorial smoking cues have elicited equivalent levels of craving from male and female smokers [9,17]. We also found no impact of the smoking or dependence level on our outcome measures. However, our sample had limited variability in daily smoking and nicotine dependence. Thus, recruiting smokers representing a wide range of smoking levels in future studies might allow differences due to daily smoking or nicotine dependence levels to emerge.

## 5. Conclusions

Overall, the present study supported the feasibility of combining three different personal smoking and three different personal nonsmoking cues in a lab-based cue reactivity paradigm and revealed that exposure to these personalized cue combinations led to strong cue-induced craving and smoking behavior differences as a function of smoking cues. Deprivation prior to cue exposure led to additive effects on the craving to smoke and smoking behavior but was not interactive. The sample size for this pilot study was small, and the design disallowed for the examination of the reactivity to each component cue. However, robust craving and smoking behavior responses to personal tri-cue combinations were revealed. The evaluation of the potential efficacy of new treatments and techniques aiming to reduce cue responding and enhance addiction treatments (medications, brain stimulation, cognitive tasks) may be better revealed under conditions that mimic probable real-world cue exposures, such as the presence of combined environment, people, and proximal stimuli that elicit strong craving and promote subsequent smoking. This study offers a methodology for capturing individuals’ most salient smoking cues in the laboratory, which may aid in future efforts to better understand and reduce this unrelenting source of smoking maintenance and relapse risk among smokers.

## Figures and Tables

**Figure 1 brainsci-12-01547-f001:**
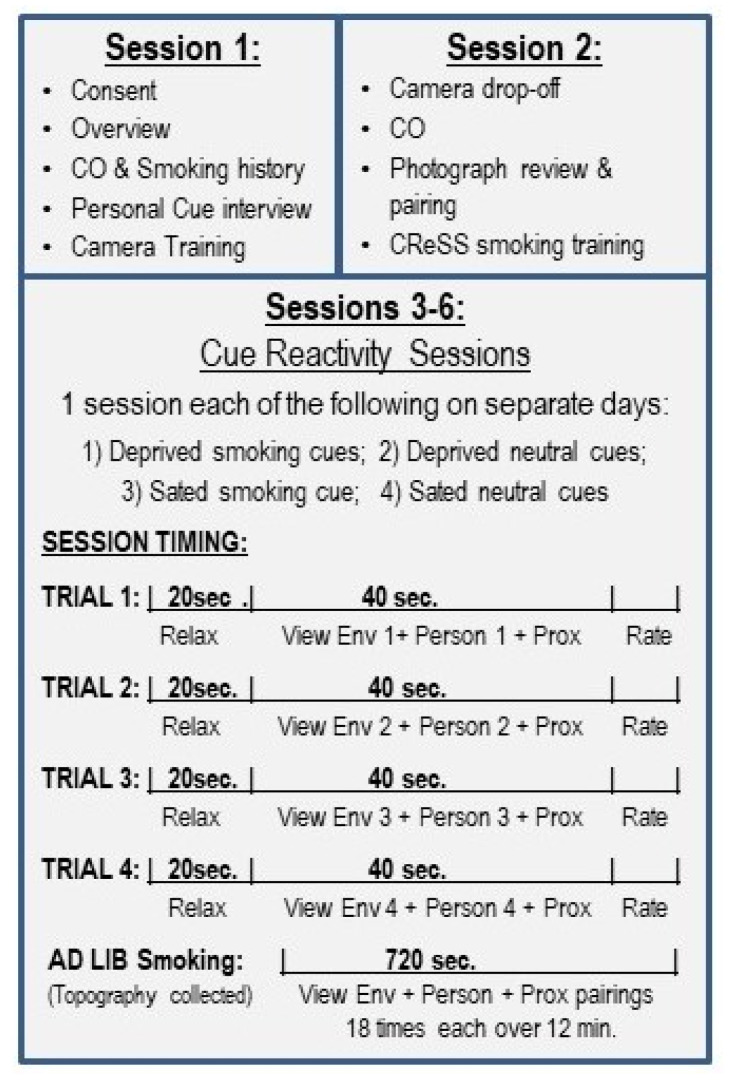
Study flow showing sessions 1–6. The cue reactivity sessions (3–6) included 4 trials during which participants relaxed (20 s), viewed combined environment (Env) + person + proximal (Prox) stimuli for 40 s in total, with 4 angles of the same environment cue alternating every 10 s and the people and proximal cues for that trial, completed post-trial craving ratings, and entered an ad lib smoking period (720 s), during which the stimuli from the 4 previous trials were repeatedly presented.

**Figure 2 brainsci-12-01547-f002:**
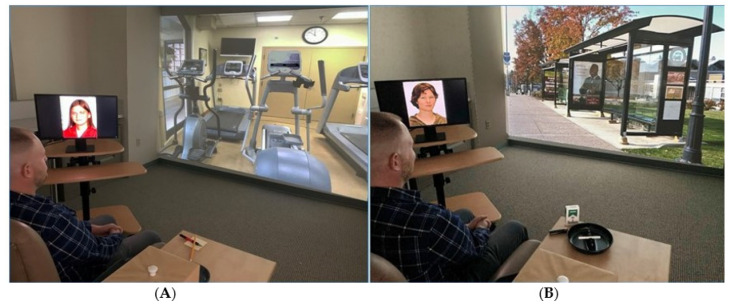
Photographs of a participant in the laboratory during cue reactivity sessions. (**A**) Nonsmoking cue session showing proximal cues (pencil and paper), environment cue (workout gym), and nonsmoking-related people cue (workout friend), and (**B**) proximal cues (participant’s cigarettes, lighter, and an ashtray), environment cue (morning bus stop), and smoking-related people cue (friend from the bus stop). In both photographs, the small box housing the CReSS smoking device is to his right, out of his visual field, until the participant is instructed, via the automated cue reactivity program, to move the tray forward, lift the box, and place one cigarette in the CReSS cigarette holder to start the ad lib smoking period.

**Figure 3 brainsci-12-01547-f003:**
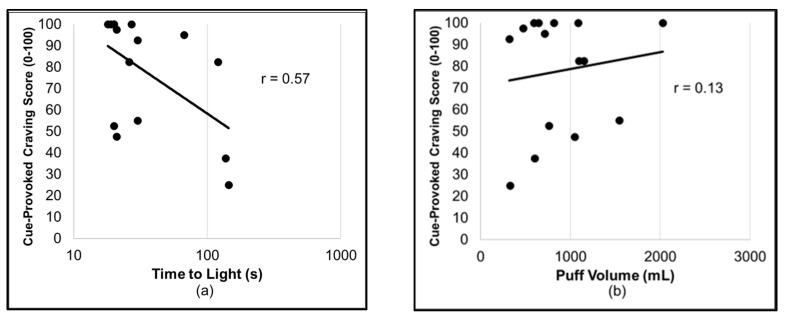
Scatter plots showing (**a**) the relationship between cue-provoked craving (0–100) and the time to light a cigarette (seconds) during the ad lib smoking period, plotted on a log scale, and (**b**) the relationship between cue-provoked craving and the total puff volume during the ad lib smoking period.

**Table 1 brainsci-12-01547-t001:** Descriptive statistics for the dependent variables.

Dependent Variable	Cue Type	Condition	
		Sated	Deprived
		Mean (SD)	Mean (SD)
Cue-Provoked Craving	Nonsmoking Cues	9 (14)	53 (34)
	Smoking Cues	30 (29)	78 (27)
Latency to Light (s)	Nonsmoking Cues	578 (264)	77 (177)
	Smoking Cues	381 (352)	24 (37)
Puff Volume (mL)	Nonsmoking Cues	130.3 (251.6)	649.2 (473.9)
	Smoking Cues	394.3 (440.8)	873.3 (447.8)

## Data Availability

Data supporting the reported results can be obtained from the corresponding author, Cynthia A. Conklin, conkca@upmc.edu.

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
