# Peer review of "Smokers’ Self-Report and Behavioral Reactivity to Combined Personal Smoking Cues (Proximal + Environment + People): A Pilot Study"

_brainsci, 2022, doi:10.3390/brainsci12111547_

Round 1

Reviewer 1 Report

The study tried to use multi-cue combination to investigate their effect on the craving rating in satied and deprived smokers. However, I think there were a few major points need address.

1) The study has a very limited sample of 15 participants. There was a large individual difference so 15 would be a small number. The authors must provide rational of their judgement of the sample size (e.g., power analysis).

2) The purpose of this study was not well documented. If the authors would like to demonstrate their method was better than traditional single stimulus cue reactivity task, then, they missed the traditional methods as a baseline. If they wanted to propose a new method, then they need to tone down by not saying "increase". 

3) the procedure of this experiment was very complex. I would suggest the authors to add a figure to show the experiment procedure. And if possible, provide us with photo with a real-experiment setting.

4) I am concerning the measure time to light. as illustrated in figure 1, this variable had a mean of about 50-70, which does not correspond to the data presented in table 1. Also, it was measured with ms, 100ms was very quick even in behavioral experiments.

Author Response

Reviewer 1

The study tried to use multi-cue combination to investigate their effect on the craving rating in satied and deprived smokers. However, I think there were a few major points need address.

1) The study has a very limited sample of 15 participants. There was a large individual difference so 15 would be a small number. The authors must provide rational of their judgement of the sample size (e.g., power analysis).

We based power for this pilot study on the cue-induced craving between the 2 smoking cue versus two nonsmoking cue effect from our 2019 combined cue paper.  That effect in the prior study was d=1.08. Thus, we calculated 97.3% power with 15 subjects in this study a p<.05. We now include this information in the revised manuscript.  

2) The purpose of this study was not well documented. If the authors would like to demonstrate their method was better than traditional single stimulus cue reactivity task, then, they missed the traditional methods as a baseline. If they wanted to propose a new method, then they need to tone down by not saying "increase".

We agree.  The study was a pilot study to determine the feasibility of presenting 3 cue types, used in our past studies, simultaneously.  We have toned down the comparison to previous methods used and focused more on the feasibility and results of tri-cue presentations of personalized cues to assess cue reactivity.

3) The procedure of this experiment was very complex. I would suggest the authors to add a figure to show the experiment procedure. And if possible, provide us with photo with a real-experiment setting.

These are excellent suggestions. We now include a figure of the protocol and a sample photograph of a participant in the subject room. We also revised the procedure for readability.

4) I am concerning the measure time to light. as illustrated in figure 1, this variable had a mean of about 50-70, which does not correspond to the data presented in table 1. Also, it was measured with ms, 100ms was very quick even in behavioral experiments.

This was a typo.  Latency to light was measured in seconds and the figure is plotted on a log scale.  Thank you for pointing it out.  We have made the corrections / clarification.  

Reviewer 2 Report

Review Report

Journal: “Brain Sciences”

Type of Paper: Article

Title of the manuscript: “Personal smoking multi-cue combinations (Environment+People+Proximal) increase craving and smoking behavior in sated and deprived smokers”.

This manuscript concerns a pilot study carried out on 15 subjects with the aim of assessing the impact of the combined presentation of smoking-associated and non-smoking-associated stimuli (proximal personal, environmental, people) on subjective and behavioural reactivity among smokers in deprived and non-deprived situations. Through a protocol with various session, the authors observe the impact of combining cue types in a cue-reactivity paradigm and, based on the results obtained, emphasise the usefulness of this protocol in future experimental studies and clinical applications.

The idea of the study is interesting and certainly the topic is of great interest both for the purposes of general addiction research and its applications to people's well-being. However, the manuscript shows several critical points which I will list point by point below. 

TITLE : Since this is a pilot study with a minimum sample size of 15 subjects, I would reformulate the title by inserting the specification that it is a “pilot study”.

ABSTRACT: I would insert that the manuscript concerns a “pilot study” also in the abstract.

1.       Introduction.

The introduction should be improved to make it easier to read and, above all, more references should be added to the statements made, note that the references used are not very recent, even though there are several recent studies in the literature on the reaction to smoking cues.

Going into specifics:

- provide a definition of cue and cue reactivity (with reference) as the main object of the study which must be understood by the reader for a good enjoyment of the text. I highlight for example the recent meta-analysis

Betts, J. M., Dowd, A. N., Forney, M., Hetelekides, E., & Tiffany, S. T. (2021). A meta-analysis of cue reactivity in tobacco cigarette smokers. Nicotine and Tobacco Research, 23(2), 249-258. https://doi.org/10.1093/ntr/ntaa147

- lines 34-36: provide a reference on the definition of "smoking cue" given here

- lines 39-42: the authors have rightly referred to previous work by their group. However please including other studies, if any, that have dealt with the reaction to distal smoking cues; the reference for studies on "...changes in mood..." should also be specified (line 42).

- Lines 45-48 provide references for more support of the statements

2.       Material and Methods.

Participants: provide literature for selection of smokers of 10 or more cigarettes (lines 89-90) and for CO concentration (lines 90-91).

Sessions: the description of the sessions is not very clear (are there 4 or 6?) and is verbose in places. For greater clarity it would be useful to have a figure/scheme with a summary of the protocol and examples of the images administered as stimuli. I would also introduce section 2.2 'experimental protocol' with a brief summary and then proceed with the details of the different experimental sessions. 

- Lines 100-101 : provide the reference for "...semi-structured interview ...in our past work".

Data Analysis: define which are the dependent variables (line 192), remind for clarity which are the Group demographics (line 195).

3. Results

- Table 1: indicate in more detail the subject of the columns then the conditions (sated, deprived); the dependent variables because at present the table is not very clear.

- Figure 1: indicate the unit of measurement in the Y-axis and also specify in the legend.

3. Discussion

More references to international literature should be included to support the statements made:

e.g. lines 252 " cue reactivity is a well-established phenomenon".

e.g. lines 257-258 "As noted in even the earliest conditioning studies".

e.g. line 261; "our prior work" cite which

- Lines 309 : "future research, peraphs using eye-tracking methods": indicate previous studies that have used this technique in similar domains (addictions) e.g. Rahmani, N., Chung, J., Eizenman, M., Jiang, P., Zhang, H., Selby, P., & Zawertailo, L. (2022) Differences in attentional bias to smoking-related, affective, and sensation-seeking cues between smokers and non-smokers: an eye-tracking study. Psychopharmacology, 1-11. https://doi.org/10.1007/s00213-022-06245-y or others as the authors deem appropriate.

- Lines 310-311: regarding the lack of impact of gender and smoking level on cue reactivity, in addition to supporting the result by referring to studies by the authors (to be included, reference missing) and others (Karelitz, 2019), studies showing the opposite, i.e. an increased reaction to conditioned stimuli to smoking in women in both behavioural studies Kenneth A. Perkins, Debra Gerlach, Josh Vender, Jennifer Meeker, Shari Hutchison, James Grobe, Sex differences in the subjective and reinforcing effects of visual and olfactory cigarette smoke stimuli, Nicotine & Tobacco Research, Volume 3, Issue 2, May 2001, Pages 141-150, https://doi.org/10.1080/14622200110043059 and in neurophysiological studies Inguscio, B. M., Cartocci, G., Modica, E., Rossi, D., Martinez-Levy, A. C., Cherubino, P., ... & Babiloni, F. (2021). Smoke signals: A study of the neurophysiological reaction of smokers and non-smokers to smoking cues inserted into antismoking public service announcements. International Journal of Psychophysiology, 167, 22-29. https://doi.org/10.1016/j.ijpsycho.2021.06.010

4. Conclusions

The authors should better specify the limitations of the study

Author Response

Reviewer 2

This manuscript concerns a pilot study carried out on 15 subjects with the aim of assessing the impact of the combined presentation of smoking-associated and non-smoking-associated stimuli (proximal personal, environmental, people) on subjective and behavioural reactivity among smokers in deprived and non-deprived situations. Through a protocol with various session, the authors observe the impact of combining cue types in a cue-reactivity paradigm and, based on the results obtained, emphasise the usefulness of this protocol in future experimental studies and clinical applications.

The idea of the study is interesting and certainly the topic is of great interest both for the purposes of general addiction research and its applications to people's well-being. However, the manuscript shows several critical points which I will list point by point below.

TITLE : Since this is a pilot study with a minimum sample size of 15 subjects, I would reformulate the title by inserting the specification that it is a “pilot study”.

We agree and have changed the title to reflect that it was a pilot study

ABSTRACT: I would insert that the manuscript concerns a “pilot study” also in the abstract.

Yes, we have done so.

  1. Introduction.

The introduction should be improved to make it easier to read and, above all, more references should be added to the statements made, note that the references used are not very recent, even though there are several recent studies in the literature on the reaction to smoking cues.

We have edited the prose to make it more readable and now offer several newer citations in support of the topics raised in the Introduction (and throughout the paper).

Going into specifics:

- provide a definition of cue and cue reactivity (with reference) as the main object of the study which must be understood by the reader for a good enjoyment of the text. I highlight for example the recent meta-analysis

Betts, J. M., Dowd, A. N., Forney, M., Hetelekides, E., & Tiffany, S. T. (2021). A meta-analysis of cue reactivity in tobacco cigarette smokers. Nicotine and Tobacco Research, 23(2), 249-258. https://doi.org/10.1093/ntr/ntaa147

We now cite this more recent paper. Additionally, we now define cue and cue reactivity with relevant citations.

- lines 34-36: provide a reference on the definition of "smoking cue" given here

We now define this and offer a relevant citation.

- lines 39-42: the authors have rightly referred to previous work by their group. However please including other studies, if any, that have dealt with the reaction to distal smoking cues; the reference for studies on "...changes in mood..." should also be specified (line 42).

We now offer additional references from other laboratories as well as relevant additional references from our past work.

- Lines 45-48 provide references for more support of the statements

This has been done throughout.

  1. Material and Methods.

Participants: provide literature for selection of smokers of 10 or more cigarettes (lines 89-90) and for CO concentration (lines 90-91).

We are not sure what the reviewer means by offering support for smoking 10 or more cigarettes per day. That was the cutoff we determined based on rates of U.S. smoking and to ensure getting regular smokers. We now state this and cite:  Vital Signs: Current Cigarette Smoking Among Adults Aged ≥18 Years --- United States, 2005—2010. Weekly September 9, 2011 / 60(35);1207-1212. We also offer support for why we chose 8ppm as the CO cutoff.  Determination of smoking cessation is thought to best be reflected in CO levels <8.  We chose this to ensure that those who entered the study were regular smokers.  We cite: Perkins, K. A., Karelitz, J. L., & Jao, N. C. (2013). Optimal carbon monoxide criteria to confirm 24-hr smoking abstinence. Nicotine & Tobacco Research, 15(5), 978-982. In support of this cutoff.  

Sessions: the description of the sessions is not very clear (are there 4 or 6?) and is verbose in places. For greater clarity it would be useful to have a figure/scheme with a summary of the protocol and examples of the images administered as stimuli. I would also introduce section 2.2 'experimental protocol' with a brief summary and then proceed with the details of the different experimental sessions.

This was also suggested by Reviewer 1, and we agree.  We now offer a figure with the protocol and an example picture of a participant in the laboratory during the cue presentation.  We have also clarified the methods to be more straightforward in presenting what we did and we added a brief overview of the study.

- Lines 100-101 : provide the reference for "...semi-structured interview ...in our past work".

We now include this reference.

Data Analysis: define which are the dependent variables (line 192), remind for clarity which are the Group demographics (line 195).

We now define the DVs and specifiy the group demographics (FTND, Cigarettes per day, sex).

  1. Results

- Table 1: indicate in more detail the subject of the columns then the conditions (sated, deprived); the dependent variables because at present the table is not very clear.

We have clarified the Table as suggested.

- Figure 1: indicate the unit of measurement in the Y-axis and also specify in the legend.

We have clarified the graph as suggested.

  1. Discussion

More references to international literature should be included to support the statements made:

e.g. lines 252 " cue reactivity is a well-established phenomenon".

e.g. lines 257-258 "As noted in even the earliest conditioning studies".

e.g. line 261; "our prior work" cite which

- Lines 309 : "future research, peraphs using eye-tracking methods": indicate previous studies that have used this technique in similar domains (addictions) e.g. Rahmani, N., Chung, J., Eizenman, M., Jiang, P., Zhang, H., Selby, P., & Zawertailo, L. (2022) Differences in attentional bias to smoking-related, affective, and sensation-seeking cues between smokers and non-smokers: an eye-tracking study. Psychopharmacology, 1-11. https://doi.org/10.1007/s00213-022-06245-y or others as the authors deem appropriate.

- Lines 310-311: regarding the lack of impact of gender and smoking level on cue reactivity, in addition to supporting the result by referring to studies by the authors (to be included, reference missing) and others (Karelitz, 2019), studies showing the opposite, i.e. an increased reaction to conditioned stimuli to smoking in women in both behavioural studies Kenneth A. Perkins, Debra Gerlach, Josh Vender, Jennifer Meeker, Shari Hutchison, James Grobe, Sex differences in the subjective and reinforcing effects of visual and olfactory cigarette smoke stimuli, Nicotine & Tobacco Research, Volume 3, Issue 2, May 2001, Pages 141-150, https://doi.org/10.1080/14622200110043059 and in neurophysiological studies Inguscio, B. M., Cartocci, G., Modica, E., Rossi, D., Martinez-Levy, A. C., Cherubino, P., ... & Babiloni, F. (2021). Smoke signals: A study of the neurophysiological reaction of smokers and non-smokers to smoking cues inserted into antismoking public service announcements. International Journal of Psychophysiology, 167, 22-29. https://doi.org/10.1016/j.ijpsycho.2021.06.010

We have revised all of these areas to include more relevant citations from the literature, including more international studies, and have offered evidence from studies suggesting alternative outcomes.

  1. Conclusions

The authors should better specify the limitations of the study

We have revised the manuscript to elaborate on the limitations and have included a line about limitations in the conclusion paragraph.

Reviewer 3 Report

This manuscript reports on the results of a pilot study of the feasibility of using combined cues (personal proximal + environment + people) in a cue reactivity study in smokers.  Clear results support the expectations of higher craving with exposure to combined smoking cues vs. combined non-smoking cues, as well as higher craving in a deprived vs. sated state.  I have only a few specific comments.

COMMENTS:

-       There is some need for general language editing

 -       There is clear evidence that the combined smoking cues elicit greater craving than the combined non-smoking cues.  The authors recognize a limitation of their pilot study, though: is the cue reactivity elicited by the combined smoking cues greater than what would be observed for the component cues alone?  Are there other published data that the authors can point to that would make the case that combined cues produce stronger craving and reactivity?  If not, then there may not be a need to undertake the effort of creating the combined cue scenarios. 

- Alternatively, there may be different experimental questions that lend themselves to studying combined cues vs. the cues independently.  The authors do touch on this in their discussion regarding the value of using combined smoking cues, but they could discuss scenarios/experimental questions where it would be sufficient or of value to study the individual component cues.

Author Response

Reviewer 3

his manuscript reports on the results of a pilot study of the feasibility of using combined cues (personal proximal + environment + people) in a cue reactivity study in smokers.  Clear results support the expectations of higher craving with exposure to combined smoking cues vs. combined non-smoking cues, as well as higher craving in a deprived vs. sated state.  I have only a few specific comments.

COMMENTS:

-       There is some need for general language editing

We have edited the entire manuscript to be more straightforward and concise.  We have also included more relevant citations and have defined terms to aid readability.

 -       There is clear evidence that the combined smoking cues elicit greater craving than the combined non-smoking cues.  The authors recognize a limitation of their pilot study, though: is the cue reactivity elicited by the combined smoking cues greater than what would be observed for the component cues alone?  Are there other published data that the authors can point to that would make the case that combined cues produce stronger craving and reactivity?  If not, then there may not be a need to undertake the effort of creating the combined cue scenarios. 

Yes, we did conduct a study showing that combined smoking cues led to greater reactivity than one smoking cue combined with a nonsmoking cues, or two nonsmoking cues.  We cited that in the manuscript, but have revised it to more clearly offer our rationale for wanting to examine three smoking cues in combination.

- Alternatively, there may be different experimental questions that lend themselves to studying combined cues vs. the cues independently.  The authors do touch on this in their discussion regarding the value of using combined smoking cues, but they could discuss scenarios/experimental questions where it would be sufficient or of value to study the individual component cues. 

We now include more details of this in the discussion section of the manuscript.

Round 2

Reviewer 1 Report

I have no more comments

Author Response

Thank you for taking to time to review our manuscript. 

Reviewer 2 Report

Compared to the revised manuscript, a few points for improvement emerge: 

1. Specify the acronym CR used in the text in the first insertion. 

2. Figure 1: Clarify better in the legend the abbreviations used in the image "Prox", "Env", "CO" etc. so that it is easier and more readily understood 

3. Line 248: "No group demographics (FTND, cigarettes per day, or sex) were correlated with the dependent variables": this statement should be in the results pragraph and not in the statistical methods

Author Response

Thank you for giving your time and consideration to reviewing our revised manuscript.  We have incorporated your suggestions as follows: 

Reviewer comments: 

1).Specify the acronym CR used in the text in the first insertion. 

We now introduce the CR acronym for cue reactivity in the first sentence. 

2. Figure 1: Clarify better in the legend the abbreviations used in the image "Prox", "Env", "CO" etc. so that it is easier and more readily understood 

We now clarify the abbreviations in the Figure legend as suggested. 

3. Line 248: "No group demographics (FTND, cigarettes per day, or sex) were correlated with the dependent variables": this statement should be in the results pragraph and not in the statistical methods

We have moved this statement to the results as recommended.